

# Shelf –Basin interaction along the Laptev – East Siberian Seas

Leif G. Anderson[1,*], Göran Björk[1], Ola Holby[2], Sara Jutterström[3], Carl Magnus Mörth[4], Matt O'Regan[4], Christof Pearce[4,5], Igor Semiletov[6,7,8], Christian Stranne[4,10], Tim Stöven[1,9], Toste Tanhua[9], Adam Ulfsbo[1,11], Martin Jakobsson[4]

[1]Department of Marine Sciences, University of Gothenburg, 412 96 Gothenburg, Sweden
[2]Department of Environmental and Energy Systems, Karlstad University, 651 88 Karlstad, Sweden
[3]IVL Swedish Environmental Research Institute, Box 530 21, 400 14 Gothenburg, Sweden
[4]Department of Geological Sciences, Stockholm University, 106 91 Stockholm, Sweden
[5]Department of Geoscience, Aarhus University, Aarhus, Denmark
[6]International Arctic Research Center, University Alaska Fairbanks, Fairbanks, AK 99775, USA
[7]Pacific Oceanological Institute, Russian Academy of Sciences Far Eastern Branch, Vladivostok 690041, Russia
[8]The National Research Tomsk Polytechnic University, Tomsk, Russia
[9]Helmholtz Centre for Ocean Research Kiel, GEOMAR, Kiel, Germany
[10]Center for Coastal and Ocean Mapping/Joint Hydrographic Center, 03824 Durham, NH, USA
[11]Division of Earth and Ocean Sciences, Nicholas School of the Environment, Duke University, Durham, NC 27704, USA

*Correspondence to*: Leif G. Anderson (leif.anderson@marine.gu.se)

**Abstract.** Extensive biogeochemical transformation of organic matter takes place in the shallow continental shelf seas of Siberia. This, in combination with brine production from sea-ice formation, results in cold bottom waters with relatively high salinity and nutrient concentrations, as well as low oxygen and pH levels. Data from the SWERUS-C3 expedition with icebreaker Oden, July to September 2014, show the distribution of such nutrient rich cold bottom waters along the continental margin from about 140 to 180 $^{\circ}$E. The water with maximum nutrient concentration, classically named the upper halocline, is absent over the Lomonosov Ridge at 140 $^{\circ}$E while it appears in the Makarov Basin at 150 $^{\circ}$E to intensify further eastwards. At the intercept between the Mendeleev Ridge and the East Siberian continental shelf slope, the nutrient maximum is still intense, but distributed across a larger depth interval. The nutrient rich water is found at salinities up to ~34.5. East of 170 $^{\circ}$E transient tracers show significantly less ventilated waters below about 150 m water depth. This likely results from a local isolation of waters over the Chukchi Abyssal Plain as the boundary current from the west is steered away from this area by the bathymetry of the Mendeleev Ridge. The water with salinities of ~34.5 has high nutrients and low oxygen concentrations as well as low pH, typically indicating decay of organic matter. A deficit in nitrate relative to phosphate suggests that this process partly occurs under hypoxia. We conclude that the high nutrient water with salinity ~34.5 are formed on the shelf slope in the Mendeleev Ridge region from interior basin water that is trapped for enough time to achieve its signature.



## 1 Introduction

The extensive, flat and shallow shelf areas of the Laptev and East Siberian seas are particularly influenced by the changing climate in the Arctic. Coastal erosion from wave action becomes widespread when the summer sea-ice cover shrinks and river discharge increases in warmer humid conditions, both affecting organic matter and nutrient supply. At the same time,

the decrease in summer sea-ice coverage changes the dynamics of the ocean by increasing vertical mixing and brine production in the fall when sea ice again starts to form over areas that in the past used to be sea-ice covered. The changes may impact shelf basin exchange (e.g. Dethleff, 2010).

Here we assess data collected in 2014, with Swedish icebreaker *Oden* during the SWERUS-C3 (Swedish – Russian – US Arctic Ocean Investigation of Climate-Cryosphere-Carbon Interactions) expedition, along the continental shelf break of

northern Siberia. SWERUS-C3 is a multi-disciplinary international program focusing on investigating the functioning of the Climate-Cryosphere-Carbon (C3) system of the East Siberian Arctic Ocean. Acquired oceanographic and bottom sediment data add to our understanding of water mass modification in the central Arctic Ocean basin. Here the objectives are to describe the spreading of shelf waters, including those richest in nutrients, from the East Siberian Sea and assess their sources, as well as to evaluate potential effects of diminishing sea-ice coverage under a warmer climate. The transient

tracers Sulphur Hexafluoride ($SF_6$) is used in this work to investigate the ventilation states of the different water masses.

The Arctic Ocean has an area of about $9.5 \times 10^{12}$ $m^2$ of which more than half is comprised of shallow continental shelf seas (Jakobsson, 2002). The deep central part consists of several basins; the Nansen and Amundsen basins are together denoted the Eurasian Basin, and the Canada and Makarov basins constitute the Amerasian Basin. The Lomonosov Ridge stretches from the continental slope of the Laptev Sea to the slope off northern Greenland and separates the Eurasian Basin from the

Amerasian Basin (Fig. 1). The deep waters of the Arctic Ocean are supplied from the Atlantic Ocean, entering either through the eastern Fram Strait (Fram Strait Branch, FSB) or over the Barents Sea (Barents Sea Branch, BSB). The latter water flows into the Kara Sea before exiting through the St Anna Trough along the continental margin where it covers a depth range down to about 1500 m (e.g., Schauer et al., 2002). Both branches flow to the east and follow the bathymetry in a cyclonic pattern around the basins (Rudels et al., 1994, Fig. 1), the difference being that the FSB takes the inner turn and is

largely restricted to the Eurasian Basin. It is mainly the BSB that flows over the Lomonosov Ridge into the Makarov Basin north of the Laptev Sea.

The upper waters are entering from both the Pacific and Atlantic Oceans, where the latter either pass over the Barents shelf or through Fram Strait. The upper waters have classically been divided into; a surface mixed layer (SML) that varies seasonally, an upper halocline of mainly Pacific origin, and a lower halocline of Atlantic origin (e.g., Jones and Anderson,

1986; Rudels et al., 1996). The flow pattern of these waters differs. The lower halocline primarily follows the underlying Atlantic layer, while the upper halocline, and even more so the surface mixed layer circulation, is much impacted by the dominating wind field (e.g. Jones et al., 2008). The flow of the surface water is dominated by a transport from the Laptev Sea towards Fram Strait, the Transpolar Drift, and one cyclonic circulation in the Canada Basin, the Beaufort Gyre. The size





of the latter is determined by the atmospheric pressure field, where a negative Arctic Oscillation results in a larger Beaufort Gyre compared to a positive Arctic Oscillation (Proshutinsky et al., 2009).

The properties of the surface mixed layer and the upper halocline are modified over the shelves, and for the SML also in the central Arctic Ocean by, e.g., mixing with river runoff, sea-ice melt and brine from sea-ice formation. Biogeochemical

processes also modify the chemical signature, e.g., lowering the nutrient concentration of the SML through primary production and increasing the nutrient concentration in the upper halocline through decay of organic matter (e.g., Jones and Anderson, 1986). The latter process has been reported to occur in the Chukchi Sea (Bates, 2006; Pipko et al., 2002), East Siberian Sea (Nishino et al., 2009; Anderson et al., 2011), and Laptev Sea (Semiletov et al., 2013, 2016).

One of the most pronounced signatures of the upper halocline of the central Arctic Ocean is a silicate maximum, which was

first reported in 1968-69 from observations made from the drifting T-3 ice island in the Canada Basin (Kinney et al., 1970). In 1979 the silicate maximum was observed during the LOREX study over the Lomonosov Ridge and into the fringe of the Amundsen Basin (Moore et al., 1983). In 1994 no silicate maximum was observed in the Makarov Basin along a section from the Chukchi Sea to the North Pole (Swift et al., 1997). It is clear that the distribution of the upper halocline with its prominent silicate signature has varied much in the past and with changing sea ice coverage it might vary even more in the

future. In this contribution we give some indications of the latter.

## 2 Methods

Water column data in this study were obtained along 6 oceanographic sections across the shelf break (A-F; Fig. 1) during the SWERUS-C3 expedition in 2014. The expedition consisted of two legs with icebreaker *Oden*. Leg 1 started 5 July in Tromsö, Norway, and followed the Siberian continental shelf to end in Barrow, Alaska, 21 August. Leg 2 took the return

route from Barrow and ended in Tromsö 3 October after concentrating the field program to the continental shelf break, slope and the adjacent deep Arctic Ocean basin. Data from Leg 2 focusing on the shelf break are discussed in this study.

Water samples were collected using a rosette system equipped with 24 bottles of Niskin type each having a volume of 7 L. The bottles were closed during the return of the CTD-rosette package from the bottom to the surface and water samples for all constituents were drawn soon after the rosette was secured in the sampling container.

The following constituents are used here; bottle practical salinity, Dissolved Inorganic Carbon (DIC), Total Alkalinity (TA), pH, oxygen, nutrients ($NO_3^- + NO_2^-$, $PO_4^{3-}$, $SiO_2$) and transient tracer $SF_6$. The order of sampling was determined by the risk of contamination meaning that transient tracer samples were collected first followed by oxygen, the carbon system parameters, nutrients and last salinity.

Salinity and temperature data were collected using a SeaBird 911+ CTD with dual SeaBird temperature (SBE 3),

conductivity (SBE 04C) and oxygen sensors (SBE 43) attached to a 24 bottle rosette for water sampling. Salinity data were calibrated against deep water samples analyzed onboard using an Autosal 8400B lab-salinometer. The salinometer was calibrated using one standard sea water ampule (IAPSO standard sea water from OSIL Environmental Instruments and Systems) before each batch of 24 samples. The accuracy of the Autosal salinities and CTD salinities should both be within





±0.003 and the accuracy for temperature ±0.002 $^{o}$C. Water samples for salinity were analysed for more than 90% of the depth and when no data were available the CTD salinity was used in the evaluation.

The water samples for determination of the transient tracer $SF_6$ were directly drawn from the Niskin bottles using 250 ml glass syringes. The samples were stored in a cooling bath that was continuously rinsed with cold surface water to prevent

outgassing of the tracers. Measurements were directly performed on board, using a purge and trap GC-ECD system similar to the "PT3" setup described in Stöven and Tanhua (2014). The column composition was as follows: The trap consisted of a 1/16" column packed with 70 cm Heysep D, the 1/8" precolumn was packed with 30 cm Porasil C and 60 cm Molsieve 5Å and the 1/8" main column with 200 cm Carbograph 1AC and 20 cm Molsieve 5Å. The precision for onboard measurements was ±0.02 fmol/kg for $SF_6$ and ±0.02 pmol/kg for CFC-12. Age modelling based on these transient tracers is complicated

and erroneous at high latitudes due to ambiguous reasons (Stöven et al., 2015; Stöven et al., 2016). Hence, we do not provide any statements about the ventilation time scale but rather the ventilation states of the water masses in the Arctic Ocean based on the concentration distribution.

An automated Winkler titration system was used for the oxygen measurements with a precision of ~1 µmol kg$^{-1}$. The accuracy was set by titrating known amounts of $KIO_3$ salts that were dissolved in sulphuric acid. As the amount was known

to better than 0.1% the accuracy should be significantly less than the precision.

DIC was determined by a coulometric titration method based on Johnson et al. (1987), having a precision of 2.0 µmol kg$^{-1}$, from duplicate sample analyses, with the accuracy set by calibration against certified reference materials (CRM, Batch #123 and #136), supplied by A. Dickson, Scripps Institution of Oceanography (USA). TA was determined by an automated open cell potentiometric titration (Haraldsson et al., 1997), with a precision better than 2.0 µmol kg$^{-1}$ and the accuracy ensured in

the same way as for DIC. pH was determined by a spectrophotometric method, based on the absorption ratio of the sulphonephtalien dye, *m*-cresol purple (mCP) (Clayton and Byrne, 1993). Purified mCP was purchased from the laboratory of Robert H. Byrne, University of South Florida, USA. The accuracy was estimated to 0.006 from internal consistency calculations of analyzed CRM samples and the precision, defined as the absolute mean difference of duplicate samples, was 0.001 pH units. The seawater pH is reported on the total scale and in situ temperature.

The partial pressure of carbon dioxide ($pCO_2$) was calculated from the combination of pH and TA and pH and DIC, using CO2SYS (van Heuven et al., 2011) with the stoichiometric dissociation constants of carbonic acid ($K_1^*$ and $K_2^*$) and bisulphate ($K_{HSO4}^*$) given by Millero (2010) and Dickson (1990), respectively. Input data included salinity, temperature, $PO_4$, and $SiO_4$ data. The reported values are the average of the two calculated for each sample. The uncertainty was computed using a Monte Carlo approach (Legge et al., 2015) and is, expressed as double standard deviation, about 2.5%.

Besides the extensive sampling and measuring of the water column, analyses were also performed on sediments. Sediment samples from 6 coring stations along the SWERUS Leg 2 cruise track (Fig. 1) were taken from 4 different depths in the upper 16 cm (Table 1). Two different types of coring devices were used; a gravity corer (GC) and multi corer (MC). These 24 samples were analysed for biogenic silica (BSi) content, with the aim of investigating a possible sedimentary source of the silicate maximum observed in the water column. Biogenic silica was measured using a wet alkaline extraction technique



(Conley and Schelske, 2002). Samples were freeze dried and approximately 30 mg of homogenized sediment was placed in a mild alkaline solution (1% $Na_2CO_3$) at 85 ºC and aliquots were taken at 3, 4 and 5 hours during this leaching process. For each of these subsamples, dissolved Si was measured by Inductively Coupled Plasma Spectrometry, using a Thermo ICAP 6500 DUO. All BSi is assumed to have dissolved after two hours leaching, after which only Si from minerals is being
released. Based on this principle, the zero-hour intercept of the slope from the 3, 4, and 5 hours Si concentrations is used to calculate the biogenic fraction. This method was validated by including blanks, and standards from a previous inter-laboratory comparison exercise (Conley, 1998). The relative uncertainties associated with this method are estimated to be ±20% of the measured value and precision of the ICP is from certified standard measurements better than 5%.

## 3 Results

The salinity distribution along the continental margin from the Lomonosov Ridge to the Chukchi Sea shows a similar general pattern, but with some significant variations especially in the top 50 meters (Fig. 2). The thinnest layer of low salinity surface water is found at the Lomonosov Ridge (section A), which increases in thickness eastward in the study area. In section B we find the lowest salinity of 24.55 at 10 m water depth, followed by a very sharp halocline with the salinity increasing from about 32 at 50 m to 34 at 100 m depth. Further to the east the halocline is less sharp with e.g., the 34 salinity
isoline deepening to a depth of more than 200 m. Here, also the > 33 isolines deepens from the shelf towards the deep basin, especially in sections D and E.

The silicate distribution is variable between the sections (Fig. 2). Over the Lomonosov Ridge (section A) the highest silicate concentration, reaching 15 µmol/L, is found in the surface. In section B the maximum is instead found at about 50 m depth and varies horizontally, with the highest concentration exceeding 30 µmol/L. At this depth the salinity is around 33. Further
to the east at section C, the concentration in the silicate maximum is higher and is found somewhat deeper and also spans over a larger salinity range. It extends horizontally all over the shelf and slope, although with concentrations decreasing some 100-150 km seaward from the shelf break. At the station farthest out in the deep basin the concentration is close to the maximum in section B.

Section D and E are fairly close to each other and both show a similar pattern. The maximum silicate concentration, above
50 µmol/L, is close to the bottom at 100-150 m depth (Fig. 2). From here the concentration decreases gradually away from the shelf break, to the lowest maximum at the outer station, around 30 µmol/L. Another specific characteristic of the silicate distribution at these sections are the wide depth range of concentrations more than 15 µmol/L. Here it spans the range of about 50 to 250 m while in section C only spans 50 to 150 m. To some degree this is attributed to the more gradual increase in salinity with depth, but there are also high concentrations at salinities above 34.5. In section F the silicate concentration is
lower and also spans a narrower depth range. However, this section starts further away from the shelf break and may be difficult to compare with the other sections.

The waters of high silicate concentration have other distinct characters such as high concentrations of the other nutrients, phosphate and nitrate, high apparent oxygen utilization (AOU) and $p$CO$_2$, and low pH (Fig. 3). The top 100-150 m is colder





than 0 °C and the nutrient maximum as represented by phosphate is largely confined to the coldest water. There are some small differences in the exact pattern of the different parameters, e.g. the AOU maximum is located slightly deeper than that of phosphate farthest out in the deep basin.

Biogenic silica concentrations in the analysed sediments varied widely between the different sites. The full names of the

5 cores include the prefix SWERUS-L2, which henceforth is omitted. The most western sites (coring stations 21MC1, 25MC1, 27MC1 ~ water column sections A, B, C) had BSi levels of less than 0.5 % (Fig. 4). Values increase slightly when going to the east, reaching up to 1% BSi in station 18MC1 (~ water column section D) and up to 2% in station 7GC1 (~ section F). Concentrations in the most eastern station 5GC1 located on the western flank of Herald Canyon are, however, much higher and reach up to 13.5%. The surface generally contains the highest concentration of BSi in all sites, except for

in station 5GC1 where the concentration increases down core. These subtler differences should however be treated with caution due to the large uncertainties associated with the measurement method.

The mean mixed layer partial pressure of SF6 along all sections is ~8.1 ppt (Fig. 5) which is slightly below the contemporaneous atmospheric value of 8.4 ppt. At all sections except A, a SF6 minimum is associated with the maximum in AOU. Close to the shelf in section B this $SF_6$ minimum is 6.4 ppt at 80 m and shoals polewards to 50 m with increasing

partial pressure to the range 6.7 - 7 ppt. The elevated AOU values are 75 - 138 μmol/kg at these depths. The SF6 minimum becomes more significant at section C with partial pressures between 4.5-5.1 ppt at 95 - 130 m (135 - 183 μmol/kg AOU). The maximum deepens eastwards to about 200 m at sections D, E and F with partial pressures between 2.5 - 3.4 ppt and 90 - 118 μmol/kg AOU.

The SF6 partial pressure in the AW layer between 250 - 600 m is homogeneously distributed with a mean value of about 6

20 ppt at section B and C (Fig. 5). In contrast, section D, E and F show significant lower mean partial pressures of 4.1 - 3.4 ppt in the same depth interval with the lowest values at section F. Note that the deep SF6 partial pressures at section E and F are close to the values in the overlying minimum at 200 m and the minimum can thus not be defined by SF6 data only. However the minima can clearly be separated by the AOU values since the warm AW layer shows constant low values of about 50 μmol/kg along all sections.

The bottom water partial pressure of SF6 has a general trend of decreasing values at a specific isobaths from the west to the east (Fig. 6). Highest partial pressures of 6.1 - 6.9 ppt can be found at section B between 100 - 500 m. Section C shows increasing partial pressures with depth from 4.6 ppt at 100 m to 6.5 ppt at 500 m, with decreasing values deeper. A similar gradient of 4.4 pt to 5.7 ppt can be found at section D, E and F at the same depth range. Below ~500 m the partial pressure decreases with increasing depth, reaching the detection limit at 1900 - 2000 m in the Makarov Basin (Fig. 6).

**4 Discussion**

The highest silicate concentrations are found at the shelf slope of sections D and E and this is also where the salinity isolines shallows (Fig. 2). This slope of the isolines infers a near bottom increase of the current due to geostrophic shear which is





superimposed on the typical overall eastward current.  The magnitude of this increase is about 3 cm/s (based on the density difference and distance between the two slope stations located at 164 m and 241 m water depth in section E) over the depth range 100-150 m which is not negligible and indicates a near bottom intensified transport of water masses in the eastward direction.  The increase in salinity along the shelf slope at bottom depths less than 250 m is accompanied by an increase in

temperature as illustrated in depth plots (Fig. 7a and 7b) with no indication of mixing with another water mass as all data from these sections have the same shape in a T-S plot (Fig 8a) for salinity > 32.5.  The maximum silicate concentration observed of about 56 µmol/L is found at ~120 m (Fig. 7c) but elevated concentrations are found down to nearly 250 m depth. Plotting silicate concentrations against salinity (Fig. 8c) shows a clear pattern with a shallow maximum around 33 and a deep maximum at 34.5.  These maxima are also evident in the section plots in Fig. 2.

When the nutrient maximum is accompanied by an oxygen minimum (Fig. 3) it suggests organic matter decay, and if this occurs at low oxygen levels, nitrate is lost by either denitrification or anammox as electron acceptor.  Such conditions can only be met close to, or in, the sediments of the shelves within the Arctic Ocean as all other waters observed in this region are well oxygenated.  A deficit in nitrate is seen when computing the property $N^{**} = 0.87 \times [NO_3] - 16 \times [PO_4] + 2.9$ (Codispoti et al., 2005), which gives a constant value if the classical Redfield-Ketchum-Richard N:P ratio (Redfield et al.,

1963) is followed.  A low value indicates denitrification.  The $N^{**}$ profiles (Fig. 7d) show a broad minimum focused at depths around 100 m, strongly indicating that this water has had its signature coloured by hypoxic conditions.

More information on the formation history of the high salinity silicate maximum water can be achieved from property versus salinity plots (Fig. 8).  The T-S curve show a typical shape for the halocline, with a warmer low salinity water at $S \approx 31$, followed by a temperature minimum at $32 < S < 33$ and then increasing temperature with salinity to a maximum in the

Atlantic Layer followed by colder water towards the highest salinity in the deep water (Fig. 8a).  The temperature minimum has historically been attributed to winter water, often with a signature of brine contribution (e.g., Aagaard et al., 1981; Anderson et al., 2013).  This brine enhanced water follows the shelf bottom and gets enriched in organic matter decay products during its flow towards the deep basin.  A nearly strait mixing line can be seen in the salinity range from about S = 34 of the temperature maximum, i.e. no $T_{min}$ at the high salinity silicate maximum water.  Oxygen profiles, on the other

hand, show a clear minimum at $S \approx 34.5$ (Fig. 8b) indicating organic matter degradation.  Comparing the oxygen signature with those of silicate and $N^{**}$ (Fig. 8 c & d) reveals interesting information.  The broad silicate maximum around $S \approx 33$ has a minimum in $N^{**}$ but no minimum in oxygen even if the concentration is some 100 µmol/kg below saturation, while the silicate maximum at $S \approx 34.5$ has a clear oxygen minimum but with only a slight minimum in $N^{**}$.  The most plausible explanation for this pattern is that the nutrient maximum at low salinity had a higher oxygen concentration before exposure

to organic matter decay at the sediment surface.  The waters with S > 32 at some stations with lower oxygen and higher silicate concentrations also have lower $N^{**}$ (Fig. 8 b, c & d), indicating less mixing and thus potentially more recent contact with the sediment surface.

The conclusion is that both nutrient maxima are formed in contact with hypoxic sediments, with one maxima at salinity around $S \approx 33$ mainly being formed on the shelf where the preformed water is well oxygenated by interaction with the





atmosphere during ice free periods, while the nutrient maximum at S ≈ 34.5 is formed at the shelf break of more than 100 m depth. Such a scenario is consistent with the $SF_6$ partial pressure of the silicate maximum at S ≈ 34.5 being close to those in the deep basin, while that around S ≈ 33 has a significantly higher level of around 7 ppt (Fig. 5). At section B the maximum AOU is associated with S ≈ 33 and found at about 50 m depth (Fig. 5b) and at C it is found at around 100 m depth (Fig. 5c).

At the latter section the maximum AOU is found at S ≈ 34.5 associated with the $SF_6$ partial pressure minimum of around 5 ppt. At the same salinity there is also a weaker minimum in section B at about 75 m depth. In section D, E, and F the $SF_6$ minimum is also found at S ≈ 34.5 but at a deeper depth of 200 m, all associated with the AOU maximum. However, at these sections the AOU maximum has a $SF_6$ partial pressure close to that of the water deeper down, indicating that the basin water in the Chukchi Abyssal Plain is the source of this high salinity nutrient maximum water. The presence of the high

salinity $SF_6$ minimum at section C and to a lesser degree at section B points to the existence of a westward penetration of water at the shelf break. However, this does not need to be a persistent flow, but can be something that occurs intermittently. The formation of the silicate maximum at S ≈ 34.5 on the shelf break is in line with Nishino et al. (2009) who suggested that the silicate maximum at this high salinity was produced by decomposition of opal-shelled organisms along the continental margin. Anderson et al. (2013) showed that this high salinity silicate maximum had a brine content of at least 4% and that

the CTD record had a small temperature minimum signature. This was not the situation in 2014, illustrating that the conditions likely are not constant with time. The sediment record (Fig 4) clearly shows that the BSi content is low in the slope off the western part of East Siberian Sea (sections B and C; coring stations 27MC1, 25MC1, 21MC1), making local decomposition in this region unlikely. Further to the east the BSi content increases slightly to the location of sections D and F, with a large increase at the most eastern station in the Herald Canyon (13.5 % BSi at site 5GC1) where opal-shelled

organism, primarily diatoms, are abundant in the bottom sediments. This strongly supports a Pacific Ocean source of silicate, but does not exclude that some of the silicate rich water enters the eastern East Siberian Sea before transformation and escape to the slope and deep central basins. However, it is not possible to fully elucidate the transport and transformation of silicate from these few sediment profiles, especially since they are also from variable bottom depths (Table 1). Nevertheless these sediment observations do not contradict occasional westward flow along the shelf break, as suggested

by the $SF_6$ signature.

Variability is also seen in a comparison with historic data. Our section F is on the border to the Chukchi Abyssal Plain, where the Arctic Ocean Section hydrographic program collected a section of data in 1994. In Fig. 9 we compare these two data sets and it is clear that the silicate maximum at S ≈ 34.5 was more or less absent in 1994. However, at stations with bottom depths ~180 m the silicate concentration was close to 20 μmol/L to the seafloor, and at the station with bottom depth

~250 m, the concentration decreased to 18 μmol/L towards the seafloor. These are the stations where the salinity does not reach the maximum salinity in Fig. 9a. Hence, there seems to be a signal from the shelf slope that did not penetrate deep into the Chukchi Abyssal Plain. Also N** had relative to the deepest data higher values at S ≈ 34 except for at the shallowest stations with elevated silicate concentrations (Fig. 9b).



Centered at S ≈ 33 the silicate maximum is higher and the N** minimum is lower in 1994, indicating a stronger contribution of organic matter decay at low oxygen levels from the shelf. There is also an indication of a wider salinity interval of the silicate maximum in 2014 compared to 1994, especially towards the high salinity end (Fig. 9a). The combination of these two features, together with observations in 2004 (Nishino et al., 2009) and 2008 (Anderson et al., 2013), suggests that more

brine has been produced on the shelves over the last decades.

Historically the extent of the nutrient maximum has varied but few studies along the Siberian continental margin have been reported. Our 2014 data show a clear maximum in the Makarov Basin, at section B some 150 nautical miles east of the Lomonosov Ridge, with higher concentrations in the sections further to the east (Fig. 2). As the concentration of silicate generally increases towards the shelf in sections B and C and also increases from section B to C (Fig 2) it is likely that the

source is the local shelf area. Data obtained from RV Polarstern in the same area as section B (see Fig. 1b for station locations) during the summer of 1996 (ACSYS 96) did not show any elevated silicate concentrations in the halocline (Fig. 10). This could be an effect of either that in 1996 the nutrient rich water was confined closer to the shelf on its transport to the east, or that the production site of this water has extended further to the west since 1996. We find it most plausible that the latter is the cause as the sea ice climate has changed significantly over the shelf south of these sections during the last 20

to 30 years (Fig. 11). Up to about the year 2000 most summers had more than 50% sea ice coverage, with a few years with less than 10%. During the last 10 years the typical conditions for the month of September is more than 90% open water. All through the record the area is more or less ice covered in November, a situation that lasts until April. Consequently, there has been more sea ice formation and, thus, brine production in this region during the last 10 years compared to the 1980's and 1990's. When this sea ice formation is further away from the coast line the initial salinity is probably also higher and

thus also the resulting brine.

In section D and F we see increasing salinity, silicate and $SF_6$ levels towards the bottom, a signature that fades away into the central deep basin (Fig. 12). This is consistent with shelf plumes penetrating down into this region. A rough computation of the fraction of shelf water can be done as follows. With an increase in $SF_6$ partial pressure from the intermediate to the bottom water of about 0.5 ppt, and the intermediate water and shelf water partial pressures being 1.5 and 8 ppt, respectively,

a little less than 10% of shelf water is needed. This is quite substantial but not unrealistic if matched with the other properties. The shelf water silicate concentration should be ~25 μmol/L in order to achieve the observed 2 μmol/L increase, and the shelf water salinity would be 36 to get an increase of 0.15. These computations completely ignore mixing and entrainment, but provide some indication that a shelf water contribution to the deep water of the Chukchi Abyssal Plain is realistic.

**5 Conclusions**

We have showed that this region of the Arctic Ocean is much more dynamic and variable than previously reported. Our data collected in the summer of 2014 is consistent with a shelf – basin exchange scenario as summarized in Fig. 13. A boundary current of Atlantic Layer water follows the shelf break from the west to the east, where some of the water crosses the



Lomonosov Ridge into the Makarov Basin. This boundary current follows the shelf break to the Alpha Ridge where it turns towards north at its western flank. The water at the corresponding depth in the Chukchi Abyssal Plain has a substantially lower partial pressure of $SF_6$, consistent with a more isolated circulation in this region.

Surface water with substantial input of river water exits the shelf north of the New Siberian Islands to follow the Lomonosov Ridge out into the central basins. High nutrient water with salinity centred at 33 exits the East Siberian Sea from its western end and contributes to the cold halocline of the central Arctic Ocean. Where the Mendeleev Ridge connects to the shelf slope a water body with salinity around 34.5, elevated nutrient concentrations and low $SF_6$ partial pressure hugs the shelf slope. Water of such property is also found further to the west. As the source of the low $SF_6$ partial pressure most likely is in the Chukchi Abyssal Plain, at least an occasional flow to the west follows, a conclusion that is supported by the surface sediment biogenic silicate (BSi) content. In the eastern study region plumes of high salinity, silicate and $SF_6$ levels flow off the shelf into the deep basin.

### Acknowledgments

We thank the supporting crew and Master of I/B Oden as well as the support of the Swedish Polar Secretariat.

This research was supported by grants from the Swedish Research Council (contract 621-2013-5105); the Swedish Research Council Formas (project reference 214-2008- 1383); the Swedish Knut and Alice Wallenberg Foundation (KAW); the European Union FP7 project CarboChange (under grant agreement no. 264879). I.S. acknowledge support from the Russian Government (grant No. 14.Z50.31.0012/03/19.2014), the Far Eastern Branch of the Russian Academy of Sciences and the ICE-ARC-EU FP7 project. The transient tracer measurements were supported by the Deutsche Forschungsgemeinschaft in the framework of the "Antarctic Research with comparative investigations in Arctic ice areas" priority program by a grant to T. Tanhua and M. Hoppema: Carbon and transient tracers dynamics: A bi-polar view on Southern Ocean eddies and the changing Arctic Ocean (TA 317=5, HO 4680=1).

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





**Table 1.** **Geographic information of sediment cores and their biogenic silica content.**

| Core ID | Longitude (°E) | Latitude (°N) | Water depth (m) | Sediment depth (cm) | Biogenic silica (wt% $SiO_2$) | Average BSi (wt%) |
|---------|---------|---------|---------|---------|---------|---------|
| L2-5-GC1 | 176.207 | 72.870 | 115.5 | 0.5 | 7.58 | 9.34 |
|  |  |  |  | 3.5 | 6.04 |  |
|  |  |  |  | 8.5 | 10.24 |  |
|  |  |  |  | 13.5 | 13.51 |  |
| L2-7-GC1 | 179.820 | 74.993 | 391.5 | 0.5 | 1.99 | 0.80 |
|  |  |  |  | 5.5 | 0 |  |
|  |  |  |  | 10.5 | 1.03 |  |
|  |  |  |  | 15.5 | 0.15 |  |
| L2-18-MC5 | 173.879 | 76.409 | 349 | 0.5 | 1.07 | 0.34 |
|  |  |  |  | 5.5 | 0.06 |  |
|  |  |  |  | 10.5 | 0.19 |  |
|  |  |  |  | 15.5 | 0.04 |  |
| L2-21-MC6 | 163.308 | 77.579 | 153 | 0.5 | 0.19 | 0.25 |
|  |  |  |  | 5.5 | 0.43 |  |
|  |  |  |  | 10.5 | 0.28 |  |
|  |  |  |  | 15.5 | 0.09 |  |
| L2-25-MC6 | 152.676 | 79.226 | 101 | 0.5 | 0.17 | 0.04 |
|  |  |  |  | 5.5 | 0 |  |
|  |  |  |  | 10.5 | 0 |  |
|  |  |  |  | 15.5 | 0 |  |
| L2-27-MC6 | 154.126 | 79.665 | 276 | 0.5 | 0.30 | 0.30 |
|  |  |  |  | 5.5 | 0.39 |  |
|  |  |  |  | 10.5 | 0.21 |  |
|  |  |  |  | 15.5 | 0.28 |  |




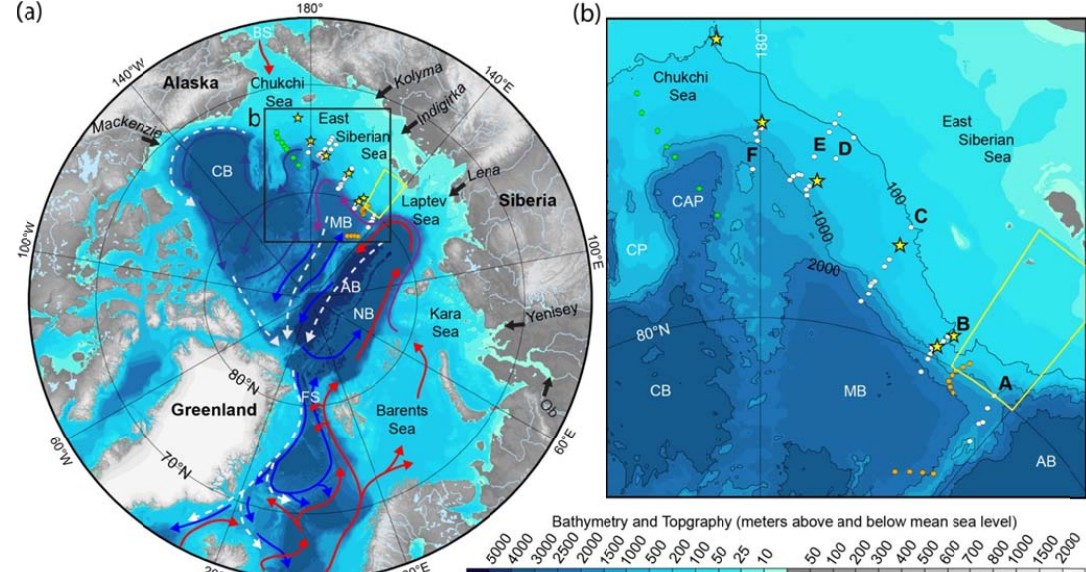

Figure 1. Map of the Arctic Ocean with general currents at intermediate depths over the deep basins and exchange with the surrounding Oceans (a). The black frame indicate the investigated area that is illustrated in b) with the hydrographic station positions of sections A to F as white points and those of sediment cores in yellow stars. The Arctic Ocean Section 1994 stations are in green and the orange points show the positions of the ACSYS 96 stations that are used as historic references. The yellow frame borders the area where the historic sea ice coverage has been evaluated, see Fig. 10. Abbreviations: Fram Strait (FS), Bering Strait (BS) Nansen Basin (NB), Amundsen Basin (AB), Makarov Basin (MB), Canada Basin (CB), Chukchi Cap (CP) and Chukchi Abyssal Plain (CAP).




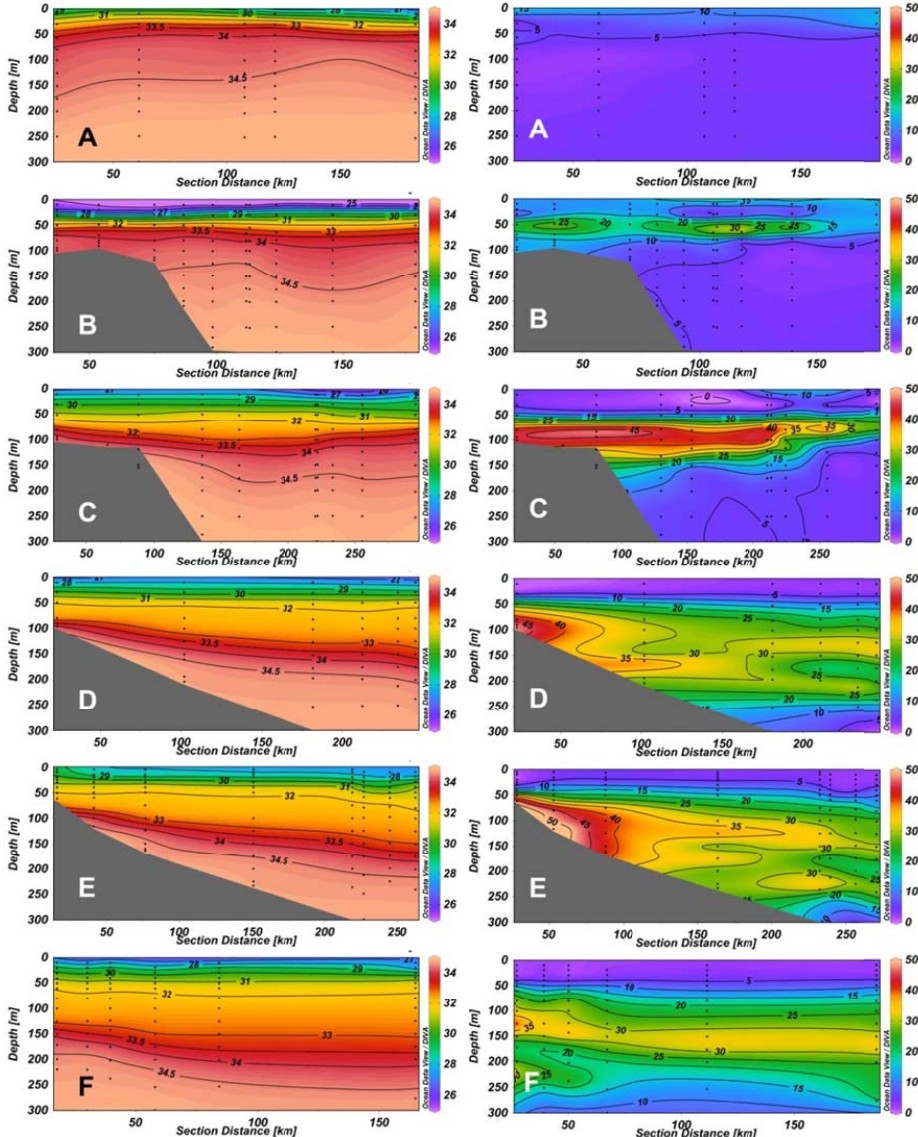

Figure 2. Sections of salinity (left) and silicate in μmol/L (right) of the upper 300 metres of sections A to F, see Fig. 1 for location of sections.




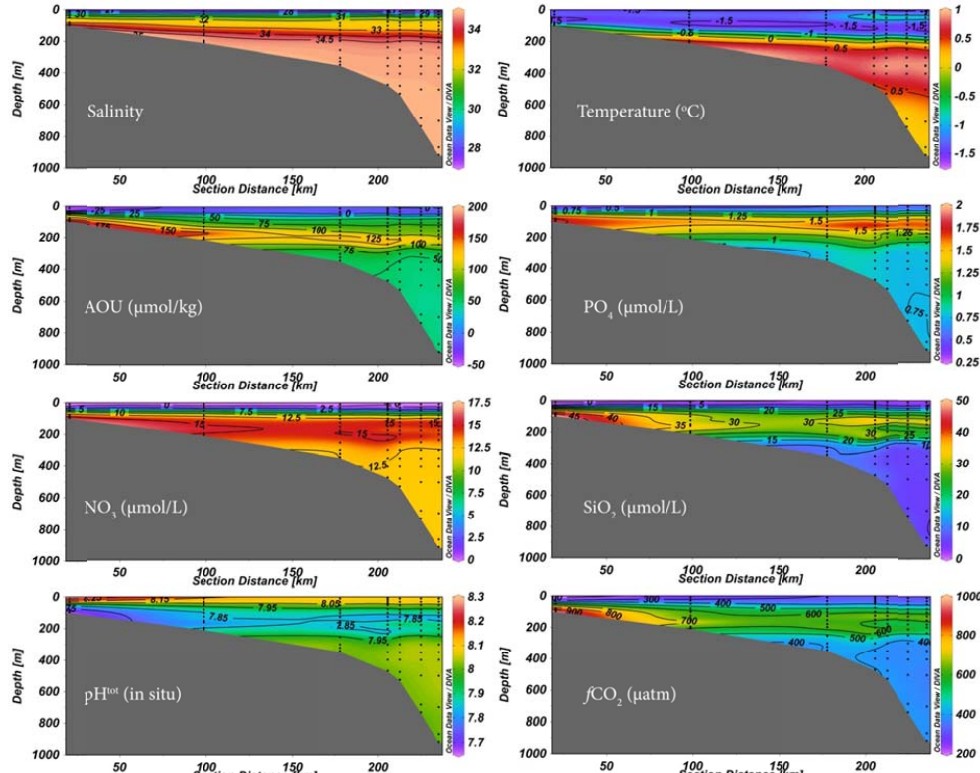

Figure 3. Properties in the upper 1000 metres along section D of Fig. 1.



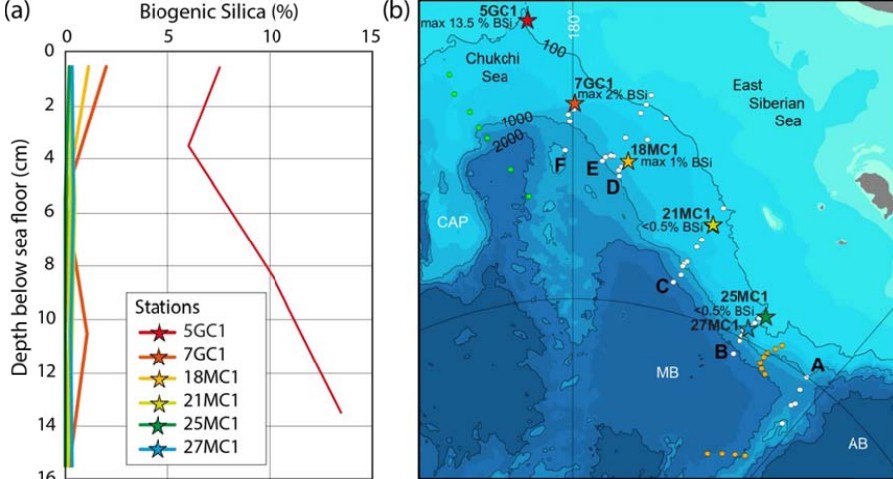

Figure 4. Biogenic silicate (BSi) concentrations in the upper 16 cm of the sediment (a) at coring sites shown in (b). The symbols of the coring sites are colour coded after measured BSi concentration; from the lowest concentrations in blue to the highest in red.





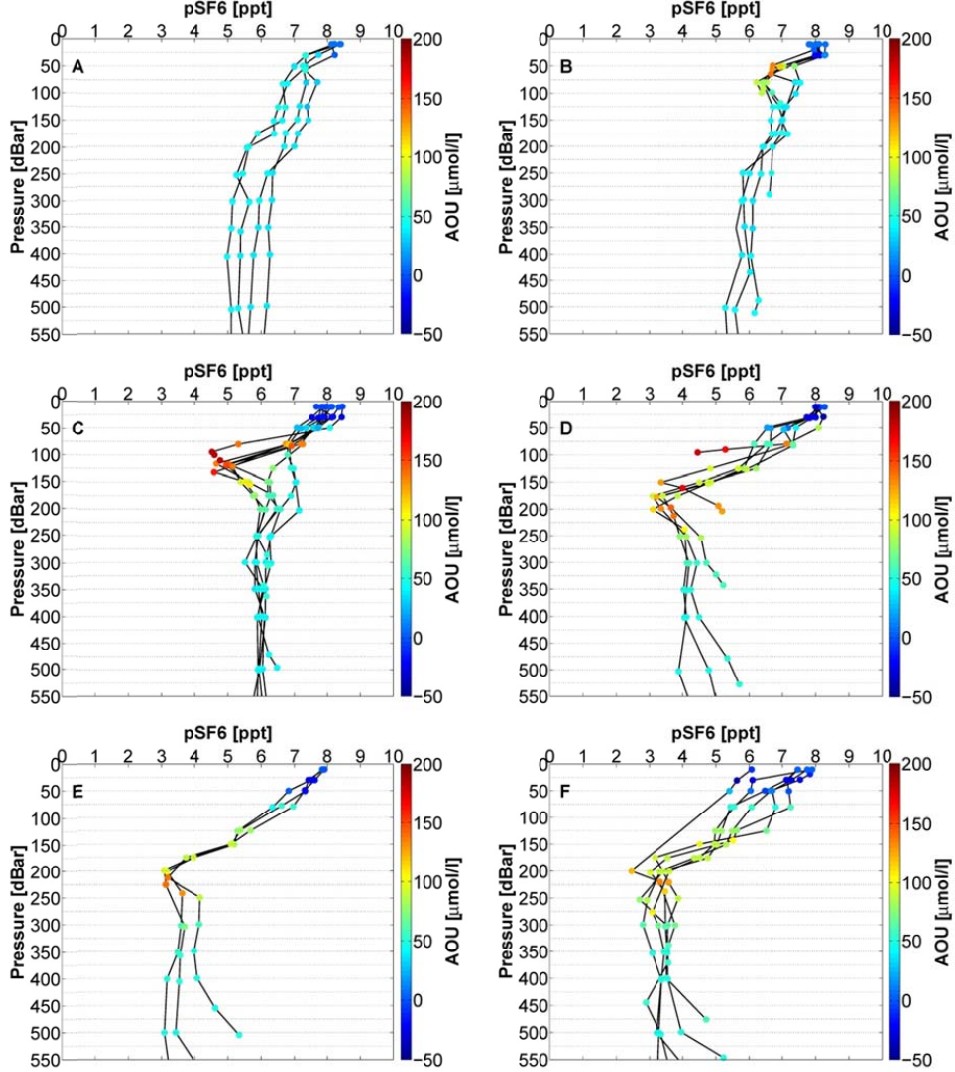

Figure 5. SF$_6$ (ppt) profiles from surface to 550 m depth, coloured by AOU (µmol/kg), of the stations in sections A to F.





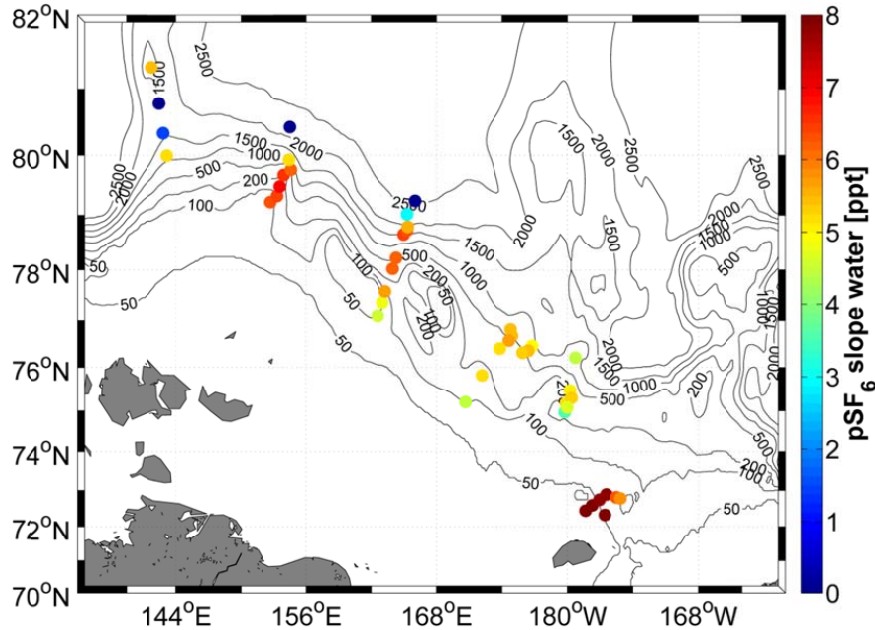

Figure 6. SF$_6$ partial pressure (ppt) in the sample collected closest to the bottom, typically 5 -10 m above.



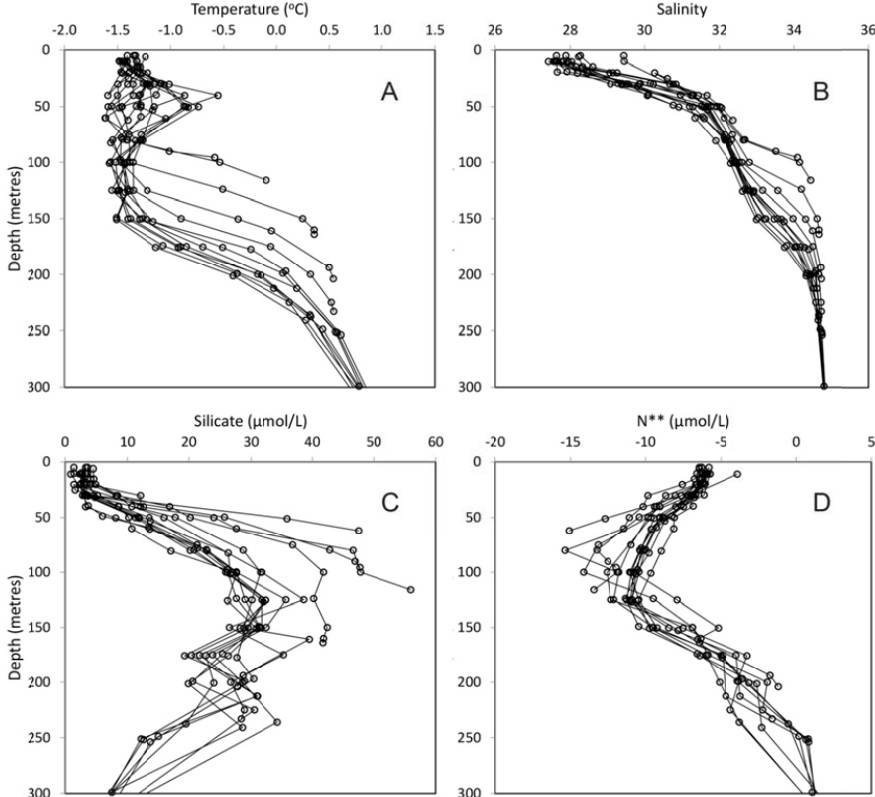

Figure 7. Depth profiles of temperature (a), salinity (b), silicate (c) and N** (d) in the upper 300 m of all stations at sections D and E, see Fig. 1 for locations.



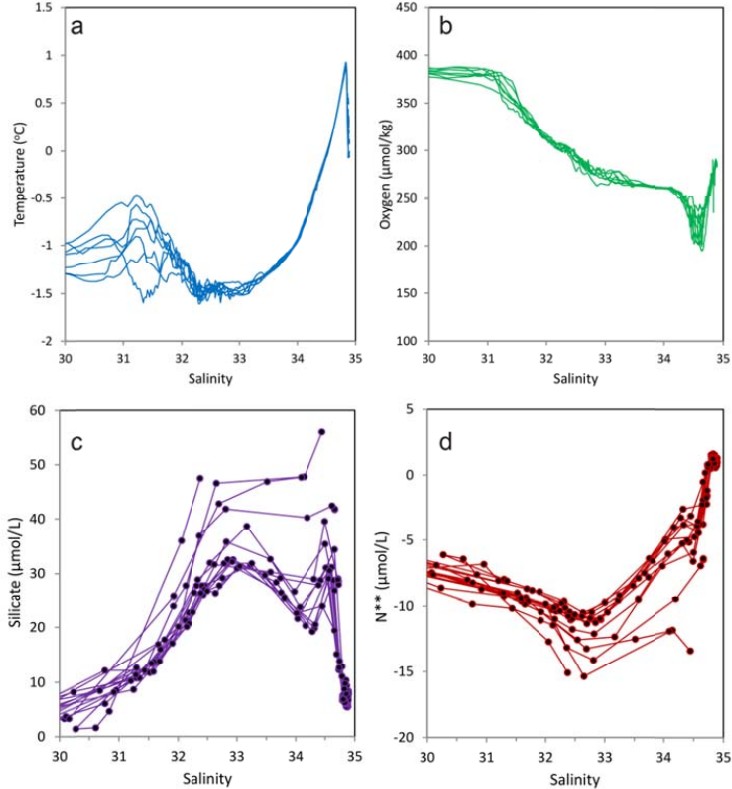

Figure 8. Temperature (a), oxygen (b), silicate (c) and N** (d) versus salinity for the stations at sections D and E. The plots
a and b are from the CTD output, while c and d are from water samples analysed.



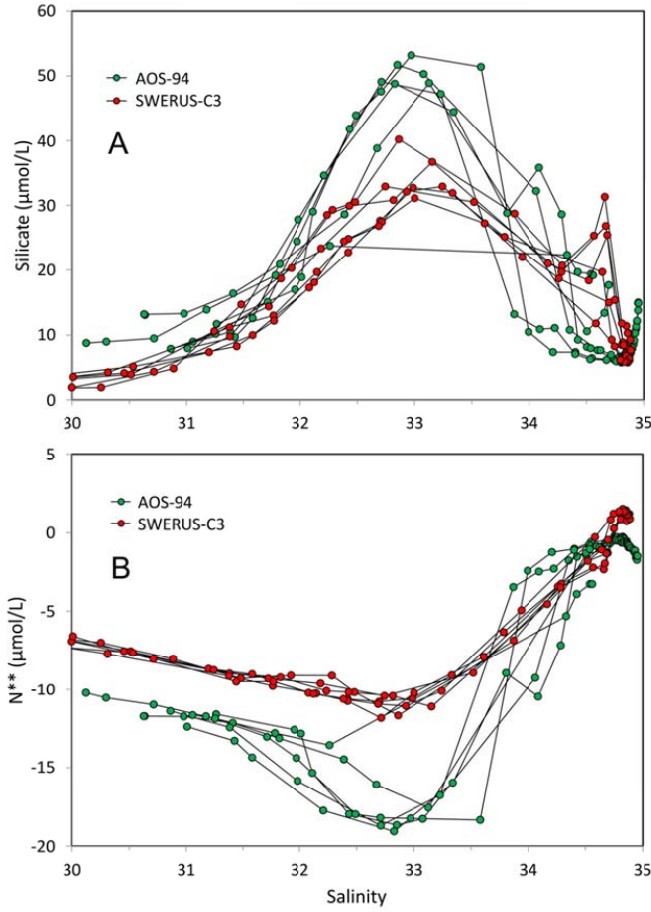

Figure 9.  Silicate (A) and N** (B) versus salinity in the Chukchi Abyssal Plain area.  Data from the Arctic Ocean Section in 1994 in green (stations along the red line in Fig. 1b) and from SWERUS-C3 in red (section F in Fig. 1b).




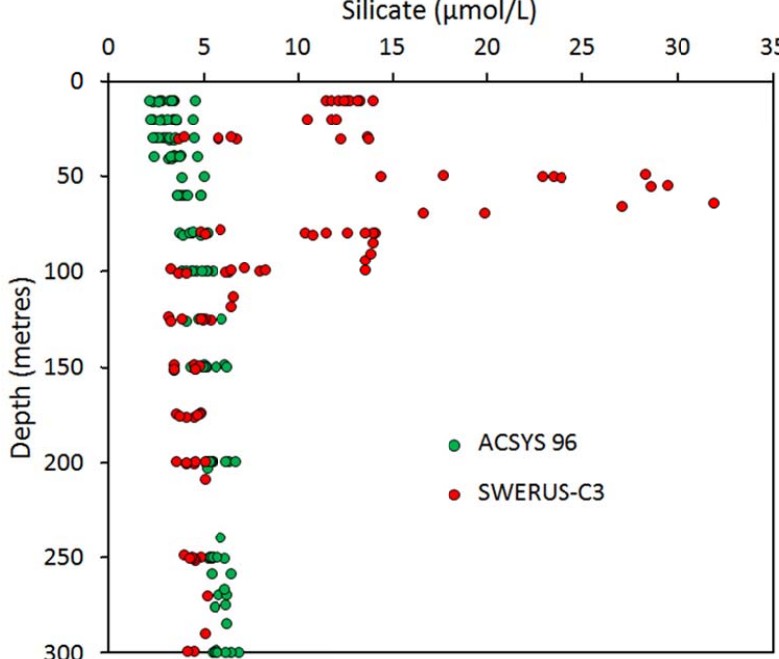

Figure 10. Silicate versus depth for data collected in 1996 (circles) and from section B in 2014 (crosses). The positions of the stations in 1996 are shown by red dots in Fig. 1b.





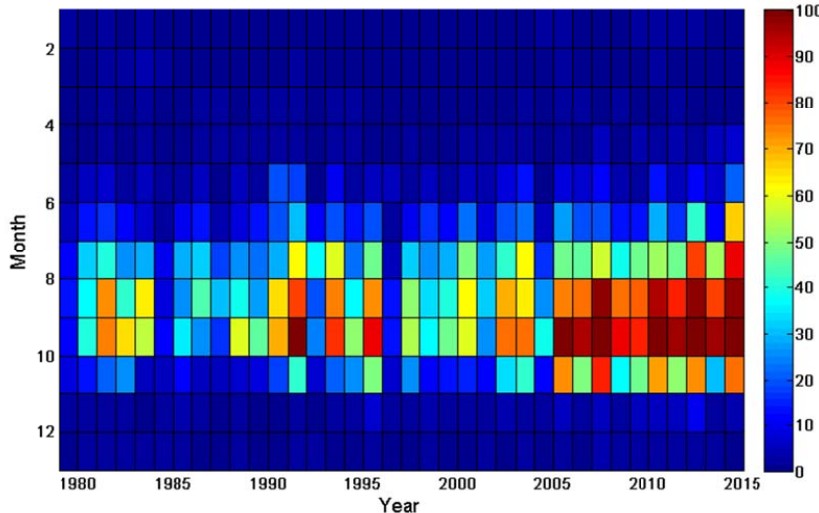

Figure 11. Percentage of ice free area in the region the region: latitude 76 to 80 °N, longitude 140 to 150 °E (framed yellow in Fig 1B), for each month from 1980 to 2014, evaluated from the passive microwave data of NSIDC (Cavalieri et al., 1996).

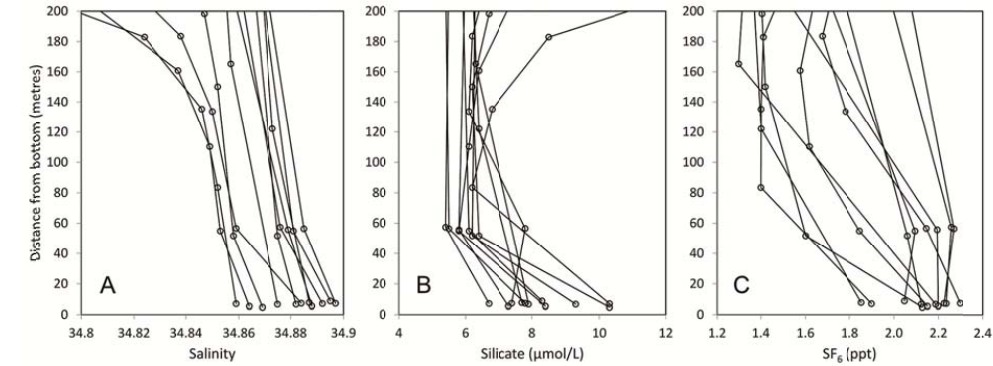

Figure 12. Salinity (A), silicate (B) and $SF_6$ (C) as a function of distance from the bottom for all stations deeper than 400 m in sections D and F. The strongest gradients in silicate and SF6 are closest to the shelf.



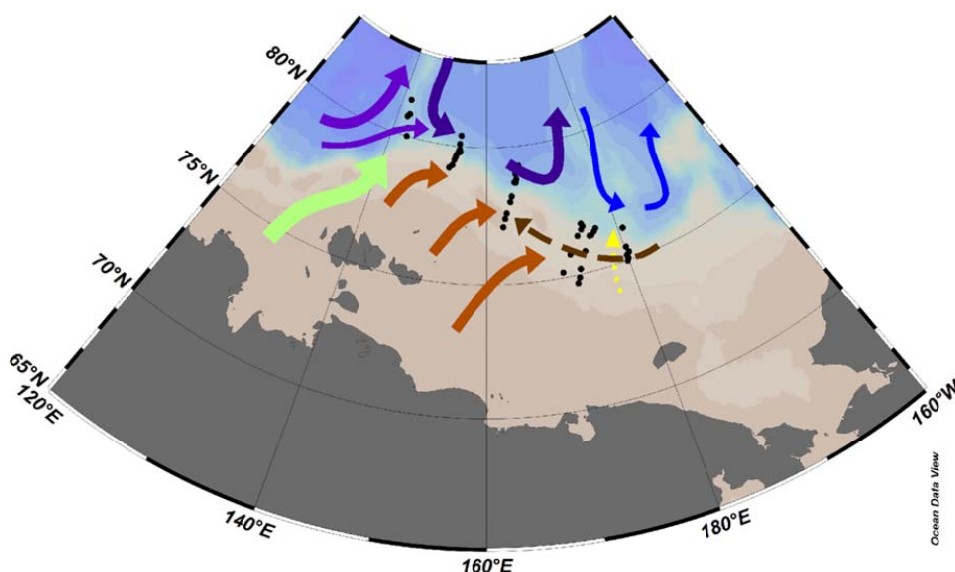

Figure 13. Summary of deduced circulation. Green arrow shows the runoff spreading in the surface out north of the New Siberian Islands. The light brown illustrates the export of nutrient rich water from the shelf into the deep basin at a salinity of around 33 and the dark brown interrupted line the nutrient rich water of a salinity around 34.5. The dark blue arrows in the deep basin show the intermediate deep (500-1500 m depth) boundary currents. The yellow dotted line illustrates the deep water plumes off the shelf break.