# Peer review of "Shelf –Basin interaction along the East Siberian Seas"

_Ocean Science, 2016_

## Referee Comment (RC1) · Anonymous Referee #1 · 5 Jan 2017

This paper uses a comprehensive hydrographic data set to assess along-slope variations in water mass properties and likely processes occurring along the ESS shelf and slope. I first applaud the authors for the development of such a comprehensive data set from a very remote and difficult portion (access-wise) of the Arctic Ocean. Second, I thought that the presentation of the various data sets was concise and the figures of high quality.

My only concern is with the estimate made on page 9 lines 25 - 30. A shelf water contribution with salinity of 36 would imply very large productions of ice in winter from a shelf region that has very low salinities (in general). Is this possible? I also wonder if the ESS has silicate concentrations of 25 umol/l. Perhaps the silicate concentrations would increase within a dense plume spreading offshore and in intimate contact with the bottom sediments? I encourage the authors to elaborate a little on both of these

points.

minor wording issues: p.5 line 21 omit "over" after "spans" p.6 line 10 remove "in" before station. p. 8 line 8. Remove "down" after water deeper.

---

## Referee Comment (RC2) · S. Nishino (Referee) · 17 Jan 2017

General comments: This paper presents analyses of hydrographic and biogeochemical data obtained from the East Siberian Arctic Seas, where only the limited data are available. Therefore, the dataset is significantly valuable. From the data, the authors proposed a new image of the Siberian shelf water spreading into the Arctic basins. Furthermore, they suggested the origin of deep silicate maximum water, which was not clear in the previous studies. This is an interesting paper that is in general clearly written and well-laid out. I recommend the paper for publication in Ocean Science after some revisions.

Specific comments: P. 1, L. 1 (Title): Did you discuss about Laptev Sea - Makarov Basin interaction with the present data? If not, it might be better to delete "Laptev"

from the title or use such as "East Siberian Arctic Seas".

P. 1, L. 25: Can you say that the water of S~34.5 is lower halocline water? If so, it might be better to add sentences to distinguish from the explanation about upper halocline water. For example, '... S~34.5, where the water is classically named lower halocline water. Here, we found new characteristics of the water ...'.

P. 2, L. 4: Probably, some references would be given.

P. 2, L. 7: Also, please see the paper below for the Makarov Basin (e.g., Figure 3 in this paper). Nishino, S., M. Itoh, W. J. Williams, and I. Semiletov (2013), Shoaling of the nutricline with an increase in near-freezing temperature water in the Makarov Basin, J. Geophys. Res. Oceans, 118, 635-649, doi:10.1029/2012JC008234.

P. 2, L. 14-15: This kind of sentences might be better to move to the Method section.

P. 3, L. 15: In addition to the introduction on the upper halocline water, it might be better to introduce some previous studies on the lower halocline water and deep silicate maximum, because this topic is another important part of the present study. You should describe more clearly what is still unknown about the deep silicate maximum. I think that the origin of the deep silicate maximum water was not clear in the previous study, but the present study sheds light on the origin from the wide-area hydrographic and biogeochemical surveys including the first SF6 measurements.

P. 4, L. 9: Is CFC-12 data used in the present study?

P. 5, L. 11-12: Is the surface low salinity with strong stratification an influence of Lena River? If possible, please explain in the discussion section.

P. 5, L. 17-18: Is the surface high silicate an influence of Lena River? If possible, please explain in the discussion section.

P. 6, L. 26 (Figure 6): It would be helpful to depict positions or a line of S=34.5 connecting each section from A to F. Or it might be better to depict SF6 distribution on the

isohaline surface of S=34.5 to identify the less ventilated area.

P. 6, L. 32: Where did you assume the reference level in the geostrophic shear calculation? Only the density field, we don't know whether the bottom currents are eastward or westward. Probably, we need discussion from current data or chemical tracer data to infer the flow direction.

P. 7, L. 4-6: I can't understand what you want to mention here. Do you want to describe implications of the bottom-intensified eastward flow?

P. 7, L. 5 (Figure 7): In Figure 7, it is not easy to understand the increase in salinity ALONG THE SHELF SLOPE with the temperature increase. Does "ALONG THE SHELF SLOPE" mean "ALONG AN ISOBATH"? If so, why didn't you show vertical sections ALONG AN ISOBATH or ALONG THE SHELF SLOPE to explain these distributions?

P. 7, L. 24: and?

P. 8, L. 1: It might be proper to describe such as "ice formation periods with cooling and convection".

P. 9, L. 5: Based on the SF6 distribution, the deep silicate maximum water (SF6 minimum and AOU maximum water) might not be related to the brine production (i.e., ventilation).

P. 9, L. 7: Figure 4 of Nishino et al. (2013; JGR) indicated the variation of nutrient maximum water along the Siberian continental margin. The variation is also associated with the recent sea ice reduction over the East Siberian Sea during the ice formation period.

P. 9, L. 15 (Figure 11): Please describe why you selected the calculation area (76-80N, 140-150E).

P. 9, L. 22: Please explain what a purpose of the analysis in Figure 12 is. Why do you

need to discuss the shelf plumes penetrating down into the central deep basin? Why does this discussion limited to the eastern part of the study area?

Same comments were inserted in the PDF file of Ocean Science Discussion.

Please also note the supplement to this comment:
http://www.ocean-sci-discuss.net/os-2016-95/os-2016-95-RC2-supplement.pdf

**Supplement:**

[revised manuscript text omitted]

---

## Referee Comment (RC3) · Anonymous Referee #3 · 14 Feb 2017

General comments

The ms is interesting, although very descriptive: some biogeochemical transformations which take place in the shallow continental shelf seas of Siberia are investigated. The ms provides useful information in order to improve the knowledge on the dynamics of shelf water, from the East Siberian Sea and assess their sources. I suggest the publication of the ms after a minor revision.

In the introduction (P.2 L.8) the authors state: "here we assess data collected in 2014" Unclear sentence is this an objective? In this case this should be reformulated. The relevance of the objectives and of the results obtained could be better stated and evidenced in the introduction and conclusion. The effect of the variations in sea-ice coverage, one of the main objectives, is not considered in the conclusion.

[Figure]

The methods used are appropriate. When describing the sediment core subsampling the authors should specify the thickness of the subsamples where the biogenic Si where analyzed. In the text umol kg-1 is used for O2 concentration but in figure 5 umol L-1 is used for AOU. Please uniform the units used throughout the ms. The transient tracers Sulphur Hexafluoride is used in this work to investigate the ventilation states of the different water masses but Overall it is unclear which benefit are derived by SF6 analysis with respect to O2, as no water mass age is derived.

For SF6 "ppt" is used as a measure units but is unclear, the authors should use International System of Units and anyway clearly indicate if the concentration is expressed on a volume basis. Is there any significant negative correlation between SF6 and AOU?

Specific comments

P.4 L.9 The precision for onboard measurements was $\pm0.02$ fmol/kg for SF6 and $\pm0.02$ pmol/kg for CFC-12: If CFC where analyzed togeteher with SF6 why this data are not presented and discussed?

P. 7. L. 16. Please clarify "indicating that this water has had its signature coloured by hypoxic conditions."

Fig. 2 Most of the capital letter are in white, a few are in black please uniform.

Fig. 3. The pH index is not readable

Fig. 4. Biogenic Silica is expressed as % dwt? This could be specified,

Fig. 11. The percentage is representing annual data and not monthly data as stated in the caption.

Fig. 12. the strongest gradients in silicates and SF6 are closest to the shul. Is unclear how the reader can verify this statement, as the sampling stations are not identifiable in the profiles-

Fig. 13. green arrows are not sufficiently evident

[Figure]

---

## Author Comment (AC1) · 28 Feb 2017

We are grateful for the referee's constructive comments as well as the kind words. Referees comments are in red.

Referee #1 My only concern is with the estimate made on page 9 lines 25 - 30. A shelf water contribution with salinity of 36 would imply very large productions of ice in winter from a shelf region that has very low salinities (in general). Is this possible? I also wonder if the ESS has silicate concentrations of 25 umol/l. Perhaps the silicate concentrations would increase within a dense plume spreading offshore and in intimate contact with the bottom sediments? I encourage the authors to elaborate a little on both of these points. Regarding the concern on high salinity and silicate we will expand this discussion with examples of some data that has been collected historically and by us.

[Figure]

minor wording issues: p.5 line 21 omit "over" after "spans" p.6 line 10 remove "in" before station. p. 8 line 8. Remove "down" after water deeper. We change the wording issues noted.

---

## Author Comment (AC2) · 28 Feb 2017

We are grateful for the referee's constructive comments as well as the kind words. Referees comments are in red.

Referee #2 General comments: This paper presents analyses of hydrographic and bio-geochemical data obtained from the East Siberian Arctic Seas, where only the limited data are available. Therefore, the dataset is significantly valuable. From the data, the authors proposed a new image of the Siberian shelf water spreading into the Arctic basins. Furthermore, they suggested the origin of deep silicate maximum water, which was not clear in the previous studies. This is an interesting paper that is in general clearly written and well-laid out. I recommend the paper for publication in Ocean Science after some revisions.

[Figure]

Specific comments: P. 1, L. 1 (Title): Did you discuss about Laptev Sea – Makarov Basin interaction with the present data? If not, it might be better to delete "Laptev" from the title or use such as "East Siberian Arctic Seas". As we don't really discuss the Laptev Sea this will be deleted from the title.

P. 1, L. 25: Can you say that the water of S~34.5 is lower halocline water? If so, it might be better to add sentences to distinguish from the explanation about upper halocline water. For example, '... S~34.5, where the water is classically named lower halocline water. Here, we found new characteristics of the water ... P. 1, L. 25: The S~34.5 water changes property along the slope thus making it complicated to name it lower halocline water. We will take a close look if it is possible to better describe this feature.

P. 2, L. 4: Probably, some references would be given. P. 2, L. 4: The following reference will be added, Charkin, A.N., Dudarev, O.V., Semiletov, I.P., Kruhmalev, A.V., Vonk, J.E., Sánchez-García, L., Karlsson, E., and Gustafsson, Ö.: Seasonal and interannual variability of sedimentation and organic matter distribution in the Buor-Khaya Gulf: the primary recipient of input from Lena River and coastal erosion in the southeast Laptev Sea. Biogeosciences, 8, 2581–941, 2011.

P. 2, L. 7: Also, please see the paper below for the Makarov Basin (e.g., Figure 3 in this paper). Nishino, S., M. Itoh, W. J. Williams, and I. Semiletov (2013), Shoaling of the nutricline with an increase in near-freezing temperature water in the Makarov Basin, J. Geophys. Res. Oceans, 118, 635-649, doi:10.1029/2012JC008234. P. 2, L. 7: We add this reference.

P. 2, L. 14-15: This kind of sentences might be better to move to the Method section. P. 2, L. 14-15: The intention was to give this information already in the introduction as a description of this contribution, but as it also is given in the methods section we delete it here.

P. 3, L. 15: In addition to the introduction on the upper halocline water, it might be better to introduce some previous studies on the lower halocline water and deep silicate maximum, because this topic is another important part of the present study. You should describe more clearly what is still unknown about the deep silicate maximum. I think that the origin of the deep silicate maximum water was not clear in the previous study, but the present study sheds light on the origin from the wide-area hydrographic and biogeochemical surveys including the first SF6 measurements. P. 3, L. 15: We appreciate this comment and will add some text along the suggestions.

P. 4, L. 9: Is CFC-12 data used in the present study? P. 4, L. 9: CFC-12 is not used in this study although it was measured simultaneously with SF6. The reason for not using CFC-12 is the decreasing atmospheric concentration in the atmosphere which causes indistinct information about ventilation due to the relative homogeneous distribution in the surface and intermediate layers. The analytical results from the CFC-12 measurements are now removed from the method section and it is now referred to the cruise report instead.

P. 5, L. 11-12: Is the surface low salinity with strong stratification an influence of Lena River? If possible, please explain in the discussion section. P. 5, L. 11-12: Yes the low salinity water over the Lomonosov Ridge is a Lena river plume signature. We will add some text on this.

P. 5, L. 17-18: Is the surface high silicate an influence of Lena River? If possible, please explain in the discussion section. P. 5, L. 17-18: And the silicate is one of its signatures, so this will also be added in the discussion part.

P. 6, L. 26 (Figure 6): It would be helpful to depict positions or a line of S=34.5 connecting each section from A to F. Or it might be better to depict SF6 distribution on the isohaline surface of S=34.5 to identify the less ventilated area. P. 6, L. 26 (Fig. 6): It would be difficult to put a S=35 line in the figure as these are the bottom water concentrations. A figure of SF6 on the S=34.5 surface is also illustrative and we suggest to include this together with a plot of S versus silicate, as suggested by referee #3, in a revised version. However Fig. 6 shows the bottom water concentrations also in the

deep basin that add to the story and thus we don't want to delete it.

P. 6, L. 32: Where did you assume the reference level in the geostrophic shear calculation? Only the density field, we don't know whether the bottom currents are eastward or westward. Probably, we need discussion from current data or chemical tracer data to infer the flow direction. P. 6, L. 32. It is a well-known fact from long term moorings that the mean current is eastward along the shelf slope. Although it is not possible to determine the absolute current velocity from just the geostrophic calculation, our data together with the known direction of the mean flow suggest that we have a bottom intensified flow in the eastward direction. We will make this clearer in the revised text.

P. 7, L. 4-6: I can't understand what you want to mention here. Do you want to describe implications of the bottom-intensified eastward flow? P. 7, L. 4-6: It illustrates that the deeper waters penetrate up on the shelf slope, which in turn have an implication for the bottom-intensified flow. We will expand on this in the text.

P. 7, L. 5 (Figure 7): In Figure 7, it is not easy to understand the increase in salinity ALONG THE SHELF SLOPE with the temperature increase. Does "ALONG THE SHELF SLOPE" mean "ALONG AN ISOBATH"? If so, why didn't you show vertical sections ALONG AN ISOBATH or ALONG THE SHELF SLOPE to explain these distributions? P. 7, L. 5 (Figure 7): We obviously used the wrong word here. The figure shows the profiles at a short longitudinal range and we should thus use the word "at" instead of "along". The intention with this figure is to show the vertical correlation between T, S, Si and N** as a complement to Fig 8 that shows the properties versus salinity. We will clarify this in the text.

P. 7, L. 24: and? P. 7, L. 24: We add "to that".

P. 8, L. 1: It might be proper to describe such as "ice formation periods with cooling and convection". P.8, L. 1: We will add text along the suggested lines.

P. 9, L. 5: Based on the SF6 distribution, the deep silicate maximum water (SF6 minimum and AOU maximum water) might not be related to the brine production (i.e., ventilation). P. 9, L. 5: This is true and we will make this point explicitly.

P. 9, L. 7: Figure 4 of Nishino et al. (2013; JGR) indicated the variation of nutrient maximum water along the Siberian continental margin. The variation is also associated with the recent sea ice reduction over the East Siberian Sea during the ice formation period. P. 9, L. 7: Figure 4 of Nishino et al 2013 covers the slope east of 175, which is the eastern part of our study. We will cite this article and look into if it adds anything to our assessment.

P. 9, L. 15 (Figure 11): Please describe why you selected the calculation area (76-80N, 140-150E). P. 9, L. 15 (Fig 11): We did this to see if there is a potential for more brine as well as organic matter production/decay in the western part of the study area as we observe the nutrient max water further to the west than earlier has been done. A statement on this will be included.

P. 9, L. 22: Please explain what a purpose of the analysis in Figure 12 is. Why do you need to discuss the shelf plumes penetrating down into the central deep basin? Why does this discussion limited to the eastern part of the study area? P. 9, L. 22: This is part of shelf-basin exchange that this contribution addresses. The signatures of plumes are only seen in the east and this information will be added to the text.

---

## Author Comment (AC3) · 28 Feb 2017

We are grateful for the referee's constructive comments as well as the kind words. Referees comments are in red.

Referee #3 General comments The ms is interesting, although very descriptive: some biogeochemical transformations which take place in the shallow continental shelf seas of Siberia are investigated. The ms provides useful information in order to improve the knowledge on the dynamics of shelf water, from the East Siberian Sea and assess their sources. I suggest the publication of the ms after a minor revision. In the introduction (P.2 L.8) the authors state: "here we assess data collected in 2014" Unclear sentence is this an objective? In this case this should be reformulated. The relevance of the objectives and of the results obtained could be better stated and evidenced in the

introduction and conclusion. The effect of the variations in sea-ice coverage, one of the main objectives, is not considered in the conclusion. P.2 L.8: The notation of the data collection is not an objective, but information. The objective is specified on lines 12-14. We delete some of the text here and expand in the method section. Also the variations of sea-ice coverage will be included in the conclusions.

The methods used are appropriate. When describing the sediment core subsampling the authors should specify the thickness of the subsamples where the biogenic Si where analyzed. In the text umol kg-1 is used for O2 concentration but in figure 5 umol L-1 is used for AOU. Please uniform the units used throughout the ms. The transient tracers Sulphur Hexafluoride is used in this work to investigate the ventilation states of the different water masses but Overall it is unclear which benefit are derived by SF6 analysis with respect to O2, as no water mass age is derived. Method section. The thickness of the sediment sub-samples will be included in Table 1. The unit of AOU in Fig 5 is wrong, should be umol kg-1 and will be changed. SF6 is used to indicate ventilation age, i.e. relative ages, which give important information with regards to the source of the S∼34.5 water with high silicate concentration. Giving ages are associated with substantial errors as we note in the method section, P. 4, L 9-10, and we thus avoid this concept.

For SF6 "ppt" is used as a measure units but is unclear, the authors should use International System of Units and anyway clearly indicate if the concentration is expressed on a volume basis. Is there any significant negative correlation between SF6 and AOU? Specific comments As for the unit of SF6 we use ppt as it avoid the issue of the temperature effect, i.e. if a water leave the surface in equilibrium with the atmosphere at different temperature it will result in different concentrations, but the same ppt. We will add a motivation on this in the method section also spelling out that it is on a volume basis. There is no correlation between AOU and SF6 if one does not restrict the data to a very close geographical and depth range. In the salinity range 34.3 to 34.7 there is an association and we suggest to include a new figure of this in a revised version.

P.4 L.9 The precision for onboard measurements was ±0.02 fmol/kg for SF6 and ±0.02 pmol/kg for CFC-12: If CFC where analyzed togeteher with SF6 why this data are not presented and discussed? P.4 L.9: CFC-12 is not used as it does not add anything and will thus be deleted from the method section.

P. 7. L. 16. Please clarify "indicating that this water has had its signature coloured by hypoxic conditions." P.7 L 16: We will clarify what we mean by the statement that the water is colored by hypoxic conditions.

Fig. 2 Most of the capital letter are in white, a few are in black please uniform. Fig. 2: We have chosen to use a color that clearly shows the letters in relation to the background color.

Fig. 3. The pH index is not readable Fig 3: We will increase the font size.

Fig. 4. Biogenic Silica is expressed as % dwt? This could be specified, Fig 4: We will specify % dwt.

Fig. 11. The percentage is representing annual data and not monthly data as stated in the caption. Fig 11: There must be some miss understanding. The color of each pixel shows the ice coverage of that month. We will clarify to avoid miss-understandings.

Fig. 12. the strongest gradients in silicates and SF6 are closest to the shul. Is unclear how the reader can verify this statement, as the sampling stations are not identifiable in the profiles- Fig 12: This is information that cannot be seen form the figure, and that's why we give this information in the legend. We can see if it is possible to modify the fig to illustrate this info.

Fig. 13. green arrows are not sufficiently evident Fig 13: We will make it a darker green.

---

## Author Response (AR1)

OS-2016-95: Comments to changes in the revised manuscript.

Referee #1

My only concern is with the estimate made on page 9 lines 25 - 30. A shelf water contribution with salinity of 36 would imply very large productions of ice in winter from a shelf region that has very low salinities (in general). Is this possible? I also wonder if the ESS has silicate concentrations of 25 umol/l. Perhaps the silicate concentrations would increase within a dense plume spreading offshore and in intimate contact with the bottom sediments? I encourage the authors to elaborate a little on both of these points.

Regarding the concern on high salinity and silicate, we have expanded this discussion with examples of observational data and modeling.

minor wording issues: p.5 line 21 omit "over" after "spans" p.6 line 10 remove "in" before station. p. 8 line 8. Remove "down" after water deeper.

We changed accordingly.

Referee #2

General comments: This paper presents analyses of hydrographic and biogeochemical data obtained from the East Siberian Arctic Seas, where only the limited data are available. Therefore, the dataset is significantly valuable. From the data, the authors proposed a new image of the Siberian shelf water spreading into the Arctic basins. Furthermore, they suggested the origin of deep silicate maximum water, which was not clear in the previous studies. This is an interesting paper that is in general clearly written and well-laid out. I recommend the paper for publication in Ocean Science after some revisions.

Specific comments: P. 1, L. 1 (Title): Did you discuss about Laptev Sea – Makarov Basin interaction with the present data? If not, it might be better to delete "Laptev" from the title or use such as "East Siberian Arctic Seas".

As aptly pointed out by the referee, we do not really discuss the Laptev Sea and this was removed from the title.

P. 1, L. 25: Can you say that the water of S~34.5 is lower halocline water? If so, it might be better to add sentences to distinguish from the explanation about upper halocline water. For example, '... S~34.5, where the water is classically named lower halocline water. Here, we found new characteristics of the water …

We added a reference for the lower halocline, but no additional comment on the new characteristics. The reason is that it is not new as the referee has observed this before and we thus leave comments regarding this to the discussion part of the text, where a short text has been included

P. 2, L. 4: Probably, some references would be given.

The following reference has been added: Charkin, A.N., Dudarev, O.V., Semiletov, I.P., Kruhmalev, A.V., Vonk, J.E., Sánchez-García, L., Karlsson, E., and Gustafsson, Ö.: Seasonal and interannual variability of sedimentation and organic matter distribution in the Buor-Khaya Gulf: the primary recipient of input from Lena River and coastal erosion in the southeast Laptev Sea. Biogeosciences, 8, 2581–941, 2011.

P. 2, L. 7: Also, please see the paper below for the Makarov Basin (e.g., Figure 3 in this paper). Nishino, S., M. Itoh, W. J. Williams, and I. Semiletov (2013), Shoaling of the nutricline with an increase in near-freezing temperature water in the Makarov Basin, J. Geophys. Res. Oceans, 118, 635-649, doi:10.1029/2012JC008234.
We have added this reference.

P. 2, L. 14-15: This kind of sentences might be better to move to the Method section.

The intention was to give this information already in the introduction as a description of this contribution, but as it also is given in the methods section we remove it here.

P. 3, L. 15: In addition to the introduction on the upper halocline water, it might be better to introduce some previous studies on the lower halocline water and deep silicate maximum, because this topic is another important part of the present study. You should describe more clearly what is still unknown about the deep silicate maximum. I think that the origin of the deep silicate maximum water was not clear in the previous study, but the present study sheds light on the origin from the wide-area hydrographic and biogeochemical surveys including the first SF6 measurements.

We appreciate this comment and have added some text along the suggestions.

P. 4, L. 9: Is CFC-12 data used in the present study?

CFC-12 is not used in this study although it was measured simultaneously with $SF_6$. The reason for not using CFC-12 is the decreasing atmospheric concentration in the atmosphere which causes indistinct information about ventilation due to the relative homogeneous distribution in the surface and intermediate layers. The analytical results from the CFC-12 measurements have been removed from the method section.

P. 5, L. 11-12: Is the surface low salinity with strong stratification an influence of Lena River? If possible, please explain in the discussion section.

Yes, the low salinity water over the Lomonosov Ridge is a Lena river plume signature. We have added some text on this in the discussion.

P. 5, L. 17-18: Is the surface high silicate an influence of Lena River? If possible, please explain in the discussion section.

See comments to the previous question.

P. 6, L. 26 (Figure 6): It would be helpful to depict positions or a line of S=34.5 connecting each section from A to F. Or it might be better to depict SF6 distribution on the isohaline surface of S=34.5 to identify the less ventilated area.

It would be difficult to put a S=35 line in the figure as these are the bottom water concentrations. A figure of SF6 on the S=34.5 surface is illustrative and we have added this together with a plot of S versus silicate, as suggested by referee #3, in this revised version. However Fig. 6 shows the bottom water concentrations also in the deep basin that add to the story and thus we do not want to remove it.

P. 6, L. 32: Where did you assume the reference level in the geostrophic shear calculation? Only the density field, we don't know whether the bottom currents are eastward or westward. Probably, we need discussion from current data or chemical tracer data to infer the flow direction.

It is a well-known fact from long term moorings and tracer data that the mean current is eastward along the shelf slope. We have added a reference to this. Furthermore we have added a sentence to make the arguments clearer in the revised text.

P. 7, L. 4-6: I can't understand what you want to mention here. Do you want to describe implications of the bottom-intensified eastward flow?

It illustrates that the deeper waters penetrate up on the shelf slope, which in turn have an implication for the bottom-intensified flow. We move this sentence before the geostrophic sheer discussion.

P. 7, L. 5 (Figure 7): In Figure 7, it is not easy to understand the increase in salinity ALONG THE SHELF SLOPE with the temperature increase. Does "ALONG THE SHELF SLOPE" mean "ALONG AN ISOBATH"? If so, why didn't you show vertical sections ALONG AN ISOBATH or ALONG THE SHELF SLOPE to explain these distributions?

We obviously used the wrong word here. The figure shows the profiles at a short longitudinal range and we should thus use the word "at" instead of "along". We have made changes and hope this is now evident.

P. 7, L. 24: and?

We add "to that".

P. 8, L. 1: It might be proper to describe such as "ice formation periods with cooling and convection".

We have added a text on this.

P. 9, L. 5: Based on the SF6 distribution, the deep silicate maximum water (SF6 minimum and AOU maximum water) might not be related to the brine production (i.e., ventilation).

This is true and we have deleted this statement.

P. 9, L. 7: Figure 4 of Nishino et al. (2013; JGR) indicated the variation of nutrient maximum water along the Siberian continental margin. The variation is also associated with the recent sea ice reduction over the East Siberian Sea during the ice formation period.

Figure 4 of Nishino et al 2013 covers the slope east of 175, which is the eastern part of our study. We added a sentence on this with a citation of Nishino et al. (2013).

P. 9, L. 15 (Figure 11): Please describe why you selected the calculation area (76-80N, 140-150E).

We did this to see if there is a potential for more brine as well as organic matter production/decay in the western part of the study area as we observe the nutrient max water further to the west than earlier has been done. We have added a short text stressing this.

P. 9, L. 22: Please explain what a purpose of the analysis in Figure 12 is. Why do you need to discuss the shelf plumes penetrating down into the central deep basin? Why does this discussion limited to the eastern part of the study area?

This is part of shelf-basin exchange that this contribution addresses. The signatures of plumes are only seen in the east and this information is added to the text.

Referee #3

General comments
The ms is interesting, although very descriptive: some biogeochemical transformations which take place in the shallow continental shelf seas of Siberia are investigated. The ms provides useful information in order to improve the knowledge on the dynamics of shelf water, from the East Siberian Sea and assess their sources. I suggest the publication of the ms after a minor revision.
In the introduction (P.2 L.8) the authors state: "here we assess data collected in 2014" Unclear sentence is this an objective? In this case this should be reformulated. The relevance of the objectives and of the results obtained could be better stated and evidenced in the introduction and conclusion. The effect of the variations in sea-ice coverage, one of the main objectives, is not considered in the conclusion.

P.2 L.8: The notation of the data collection is not an objective, but information. The objective is specified on lines 12-14. We have deleted some of the text here and expanded in the method section. Also the variations of sea-ice coverage have been included in the conclusions.

The methods used are appropriate. When describing the sediment core subsampling the authors should specify the thickness of the subsamples where the biogenic Si where analyzed. In the text umol kg-1 is used for O2 concentration but in figure 5 umol L-1 is used for AOU. Please uniform the units used throughout the ms. The transient tracers Sulphur Hexafluoride is used in this work to investigate the ventilation states of the different water masses but Overall it is unclear which benefit are derived by SF6 analysis with respect to O2, as no water mass age is derived.

Method section. The thickness of the sediment sub-samples is included in Table 1. The unit of AOU in Fig 5 is wrong, should be umol kg-1 and is changed. SF6 is used to indicate ventilation age, i.e. relative ages, which give important information with regards to the source of the S~34.5 water with high silicate concentration. Giving ages are associated with substantial errors as we note in the method section, P. 4, L 9-10, and we thus avoid this concept.

For SF6 "ppt" is used as a measure units but is unclear, the authors should use International System of Units and anyway clearly indicate if the concentration is expressed on a volume basis. Is there any significant negative correlation between SF6 and AOU?
Specific comments

As for the unit of SF6 we use ppt as it avoid the issue of the temperature effect, i.e. if a water leave the surface in equilibrium with the atmosphere at different temperature it will result in different concentrations, but the same ppt. We have added a motivation on this in the method section also spelling out that it is on a volume basis. There is no correlation between AOU and SF6 if one does not restrict the data to a very close geographical and depth range. In the salinity range 34.3 to 34.7 there is an association and we have included a new figure in the revised version.

P.4 L.9 The precision for onboard measurements was ±0.02 fmol/kg for SF6 and ±0.02 pmol/kg for CFC-12: If CFC where analyzed togeteher with SF6 why this data are not presented and discussed?

CFC-12 is not used as it does not add anything and is thus deleted from the method section (see also comment to ref #2).

P. 7. L. 16. Please clarify "indicating that this water has had its signature coloured by hypoxic conditions."
We have expanded this statement to clarify what we mean.

Fig. 2 Most of the capital letter are in white, a few are in black please uniform.
We have chosen to use a color that clearly shows the letters in relation to the background color.  We do not wish to change this if not needed for editorial reasons.

Fig. 3. The pH index is not readable
The font size has been increased.

Fig. 4. Biogenic Silica is expressed as % dwt? This could be specified,
We have specified % dwt.

Fig. 11. The percentage is representing annual data and not monthly data as stated in the caption.
There must be some missunderstanding.  The color of each pixel shows the ice coverage of that month.  We think this is clearly stated in the legend.

Fig. 12. the strongest gradients in silicates and SF6 are closest to the shul. Is unclear how the reader can verify this statement, as the sampling stations are not identifiable in the profiles-
To clarify this we have colored the stations and given the information of the relative bottom depth.  Having this information, we deleted the statement on the strongest gradients as the reader now can evaluate this.  We also modified the text somewhat in the discussion to further clarify our message.

Fig. 13. green arrows are not sufficiently evident
We have made it a darker green.

---

## Author Response (AR2)

OS-2016-95: Comments to final changes in the revised manuscript.

The sentence "The black bars represent the depth layer of the sediment that is analysed." was added at the end of figure legend 4 to clarify the symbols.